# Synthesis and Pesticidal Activity of New Niacinamide Derivatives Containing a Flexible, Chiral Chain

**DOI:** 10.3390/molecules28010047

**Published:** 2022-12-21

**Authors:** Zhe-Cheng Wei, Qiao Wang, Li-Jing Min, Joanna Bajsa-Hirschel, Charles L. Cantrell, Liang Han, Cheng-Xia Tan, Jian-Quan Weng, Yu-Xin Li, Na-Bo Sun, Stephen O. Duke, Xing-Hai Liu

**Affiliations:** 1College of Chemical Engineering, Zhejiang University of Technology, Hangzhou 310014, China; 2Key Laboratory of Vector Biology and Pathogen Control of Zhejiang Province, College of Life Science, Huzhou University, Huzhou 313000, China; 3Natural Products Utilization Research Unit, Agricultural Research Service, U.S. Department of Agriculture, Oxford, MS 38677, USA; 4State Key Laboratory of Elemento-Organic Chemistry, Department of Chemistry, Nankai University, Tianjin 300071, China; 5Key Laboratory of Study and Discovery of Small Targeted Molecules of Hunan Province, Hunan Normal University, Changsha 430081, China; 6College of Biology and Environmental Engineering, Zhejiang Shuren University, Hangzhou 310015, China; 7National Center for Natural Product Research, School of Pharmacy, University of Mississippi, Oxford, MS 38677, USA

**Keywords:** synthesis, fungicide, herbicide, molecular docking

## Abstract

Natural products are a source for pesticide or drug discovery. In order to discover lead compounds with high fungicidal or herbicidal activity, new niacinamide derivatives derived from the natural product niacinamide, containing chiral flexible chains, were designed and synthesized. Their structures were confirmed by ^1^H NMR, ^13^C NMR and HRMS analysis. The fungicidal and herbicidal activities of these compounds were tested. The fungicidal activity results demonstrated that the compound (*S*)-2-(2-chloronicotinamido)propyl-2-methylbenzoate (**3i**) exhibited good fungicidal activity (92.3% inhibition) against the plant pathogen *Botryosphaeria berengriana* at 50 μg/mL and with an EC_50_ of 6.68 ± 0.72 μg/mL, which is the same as the positive control (fluxapyroxad). Compound **3i** was not phytotoxic and could therefore be used as a fungicide on crops. Structure-activity relationships (SAR) were studied by molecular docking simulations with the succinate dehydrogenase of the fungal mitochondrial respiratory chain.

## 1. Introduction

Natural products [1,2,3] have diverse structures, high biological activities, and novel mechanisms of action. Therefore, they are a rich source for pesticide discovery [4,5,6,7]. Nicotine is a class of pyridine type natural products which was first found in tobacco [8]. Nicotine has the effects of killing insects [9] and regulating human physiological activities [10]. Nicotine is also an important intermediate [11] in the synthesis of the natural product niacin and niacinamide (Figure 1), which are essential vitamins for humans. Nicotine acid derivatives can be found as many plant or microbial secondary metabolites. For example, wilforine (Figure 1) [12], which was found as a secondary metabolite in the Chinese traditional insecticidal plant thunder god vine (*Tripterygium wilfordii*), was used to treat rheumatoid arthritis, nephritis, lupus erythematosus, and thrombocytopenic purpura.

Furthermore, the pyridine structure is also a key skeleton in many agrochemicals [13,14,15]. They include the fungicide boscalid [16], the insecticide pyrifluquinazon [17], and the herbicide fluroxypyr [18]. The excellent succinate dehydrogenase inhibitor (SDHI) boscalid (2-chloro-*N*-(4′-chloro-[1,1′-biphenyl]-2-yl)nicotinamide, Figure 2) has high activity against plant pathogens *Sclerotinia* species., *Zymospetoria* species, and *Colletotrichum* species., and it has been widely used in agriculture. It became one of the important template molecules for the development of new fungicides [19,20,21] with a niacinamide skeleton (Figure 2). In 1990, Kumagai et al. [22] discovered three microbial secondary metabolites atpenin B, atpenin A4, atpenin A5 in *Penicillium* species, which contain pyridine ring, acyl group and a longer aliphatic chain with strong inhibitory activity against SDH from bovine heart. 

In our previous work, many bioactive heterocyclic compounds were designed and synthesized [23,24,25,26]. Of these, some pyridine derivatives exhibited good herbicidal, fungicidal, and plant growth regulation activity. In this paper, based on the structure of the natural product atpenin and the fungicide boscalid, a series of chiral niacinamide derivatives were designed. 2-Chloronicotinic acid was selected as a starting material and a carboxamide group and a flexible, chiral chain was introduced (Figure 3). Niacinamide derivatives were finally synthesized, and the structures were characterized by ^1^H NMR, ^13^C NMR and HRMS analysis. The fungicidal and herbicidal activity of chiral niacinamide compounds were evaluated, and the docking studies were also carried out to study the mode of action. 

## 2. Results and Discussion

### 2.1. Synthesis and Spectra Analysis 

The synthetic route of *(S)-*2-(2-chloronicotinamido)propyl benzoates or nicotinate compounds is outlined in Figure 1. In the process of synthesizing acyl chloride 1, SOCl_2_ is not only a reactant but also a solvent, so it needs to be excess. A tail gas absorption device should be added during the reaction to avoid environmental pollution. The reaction is judged by the state of the reaction liquid, which is changed from turbid to clear, the reaction can be stopped, then excess sulfoxide chloride was removed. In the synthesis of intermediate **2**, the diluted acyl chloride 1 should be added slowly to a solution of (*S*)-2-aminopropan-1-ol, and the order cannot be changed. Furthermore, (*S*)-2-aminopropan-1-ol needs to be in excess (at least twice), which can improve the yield of intermediate **2**. When the intermediate **2** is purified by column chromatography, the by-product ((*S*)-2-aminopropyl 2-chloronicotinate) is first obtained with eluent (ethyl acetate/petroleum ether = 1/1), and the intermediate **2** is directly given with pure ethyl acetate. For target product synthesis, following the addition of one equivalent of the respective substituted benzoyl chlorides (and nicotinyl chloride **3k**), the reaction was stirred at room temperature and monitored by thin layer chromatography. 

According to the ^1^H NMR spectra of compounds **3a**~**3p**, it can be found that doublet peaks from the amide group can be found at about 6.4 ppm, with a coupling constant of 6.4 Hz. The protons of the pyridine ring and the benzene ring are located at 6–9 ppm, which are according to the aromatic ranges. The doublet peak of the chiral methyl group appearance at 1.29~1.43 ppm with coupling constant is 5.4 Hz; The CH group, which found as multiple peaks with shifts between 4–5 ppm. According to the high-resolution mass spectrometry, the difference between the measured value and the theoretical value is within 0.003.

### 2.2. Fungicidal Activity

Fungicidal activities of (*S*)-2-(2-chloronicotinamido)propyl benzoates or nicotinate compounds against *Fusarium oxysporum*, *Cercospora arachidicola*, *Botryosphaeria berengriana*, *Alternaria solani*, *Gibberella zeae*, *Sclerotinia sclerotiorum*, *Botrytis cinerea*, *Rhizoctonia solani*, *Phytophthora infestans*, *Phytophthora capsica*, and fluxapyroxad were evaluated at 50 μg/mL according to our previous work [27,28] (Table 1).

From Table 1, compounds **3a**–**3p** showed good fungicidal activity (>60%) against *B. berengriana*, which is better than that of positive control fluxapyroxad (63.6%). Among them, compounds **3i**, **3m**, **3o**, and **3p** exhibited the best activity (>80%) against *B. berengriana*, while the fungicidal activity of compound **3i** reached 92.3%, which was much higher than the control fluxapyroxad (63.6%). Against *G. zeae*, compounds **3g**, **3h**, **3j**, **3k**, **3m**, **3n** and **3p** displayed good activities (about 70%), which were better than the positive control fluxapyroxad (28.6%). The other compounds also exhibited good activity (>30%) against *G. zeae*, except compounds **3c** (21.9%) and **3d** (25.0%). For *A. solani,* compounds **3j** and **3p** exhibited moderate activity (>60%), which was lower than that of l fluxapyroxad (88.9%). Compounds **3l** and **3m** exhibited good activity (>80%) against *S. sclerotiorum*, and most of the other compounds also possessed moderate activities (>50%), but all were lower than fluxapyroxad (96.4%). Against *C. arachidicola*, compounds **3g**, **3k**, **3l**, **3m** and **3n** displayed moderate activities (>65%), but they were still lower than that of fluxapyroxad (100%). Compounds **3g**, **3h**, **3i** and **3m** displayed good activity (>57%) against *B. cinerea*, which is the same as the control (63.6%). Against *F. oxysporum*, *P. infestans*, *P. capsici*, *R. solani*, all compounds evaluated exhibited low activity.

Based on the preliminary fungicidal activity results of >80%, compounds **3i**, **3l**, **3m**, **3o**, and **3p** were selected for further study. From Table 2, compounds **3i**, **3m**, **3o** and **3p** showed excellent fungicidal activity against *B. berengriana* with EC_50_ values of 6.68 ± 0.72, 15.14 ± 1.21, 11.98 ± 1.04 and 7.93 ± 1.05 μg mL^−1^, respectively, which are similar to fluxapyroxad (7.93 ± 1.05 μg mL^−1^). Among these compounds*,* compound **3i** (6.68 ± 0.72 μg mL^−1^) exhibited the best activity, which is better than fluxapyroxad (7.93 ± 1.05 μg mL^−1^). For *S. sclerotiorum,* compounds **3l** and **3m** exhibited good activity with EC_50_ values of 12.69 ± 1.15 and 11.83 ± 1.03 μg mL^−1^, respectively, which were lower than that of fluxapyroxad (0.73 ± 0.11 μg mL^−1^).

### 2.3. Phytotoxicity 

The phytotoxicity of *(S)-*2-(2-chloronicotinamido)propyl benzoates or nicotinate compounds are listed in Table 3. As shown in Table 3, most of the *(S)-*2-(2-chloronicotinamido)propyl benzoates or nicotinate compounds exhibited low phytotoxicity against lettuce (*Lactuca sativa*) and bentgrass (*Agrostis stolonifera*) at 1 mM, except compounds **3l** and **3p**. Compounds **3l** and **3p** possessed moderate phytotoxic (ranking 3) against *Agrostis stolonifera* at 1 mM. Compounds **3c**, **3d**, **3e**, **3h**, **3m**, **3n** possessed weak phytotoxicity (ranking 1~2) against *Agrostis stolonifera* at 1 mM. For the dicot lettuce, most of these compounds exhibited low weak phytotoxicity (ranking 1~2) against *Agrostis stolonifera* at 1 mM. The commercial herbicide aminotriazole was more active than any of the synthesized compounds on both lettuce and bentgrass.

### 2.4. Molecular Docking Studies 

The niacinamide compounds in this study were designed from reported SDH inhibitors. The fungicidal activity of **3i** is good, so we conducted molecular modeling analysis of binding of this compound with SDH. The molecular docking was done and results are shown in Table 4 and Figure 4. From Table 4, the docking scores of compound **3i** and the positive control, fluxapyroxad, are similar. As shown in Figure 4A, there are two hydrogen bonds between Trp173 or Tyr58 and the O atom of CONH group with the distances of 1.9 Å and 2.1 Å, respectively. On the other hand, there were three π-cation interactions between Arg43 and the phenyl ring with the distances of 3.8 Å, 4.1 Å, 4.7 Å. Furthermore, there was a π-π interaction between the pyridine ring and Tyr58 with a distance of 4.8 Å. As for the positive control fluxapyroxad (Figure 4B), it had a different molecular interaction mode with compound **3i**, although it also has two hydrogen bonds, three π-cation interactions and a π-π interaction with the same amino acid residue. The π–cation interactions were between the pyrazole ring and the Arg43, and the hydrogen bonds exist between the NH and Trp173 or Tyr58. This may be the reason of different fungicidal activities between **3i** and fluxapyroxad. 

## 3. Materials and Methods

### 3.1. Instruments 

Melting points were determined using an X-4 apparatus and uncorrected. ^1^H NMR and ^13^C NMR spectra were measured on a Bruker AC-P500 and AC-P400 instrument using TMS as an internal standard and CDCl_3_ as the solvent. HR-ESI-MS was tested using an Agilent 1100 HPLC-JEOL AccuTOF instrument. All reagents were of analytical grade or freshly prepared before use.

### 3.2. Synthesis 

#### 3.2.1. Synthesis of Intermediate **2**

The synthetic route is shown in Figure 1. 2-Chloronicotinic acid (3.4 g, 22 mmol) and SOCl_2_ (10 mL) were heated at reflux for 3h, until the reaction solution becomes clear, then reflux for another 30min, evaporated SOCl_2_ to obtain yellow liquid **1** without purification. Then, the solution of 2-chloronicotinoyl chloride (14.1 g, 0.08mol) in THF (10 mL) was added in a dropwise fashion to the mixture of (*S*)-2-aminopropan-1-ol (7.5g, 0.1 mol) and triethylamine (8 mL) in THF (20 mL), the mixture was stirred for 4 h. The solvent was removed, and pure product of intermediate **2** was given as a white solid by column chromatography. White solid, m.p.62–64 °C, ^1^H NMR (CDCl_3_, 500 MHz), δ: 1.25 (d, *J* = 5.5 Hz, 3H, CH_3_), 3.35 (t, *J* = 4.5 Hz, 1H, OH), 3.57–3.62 (m, 1H, CH_2_), 3.69–3.73 (m, 1H, CH_2_), 4.18–4.23 (m, 1H, CH), 6.93 (d, *J* = 6.1 Hz, 1H, NH), 7.27–7.29 (m, 1H, Py), 7.92–7.94 (m, 1H, Py), 8.37–8.39 (m, 1H, Py).

#### 3.2.2. Synthesis of Target Compounds **3**

The synthetic route is shown in Figure 1. To a solution of intermediate **2** (0.3 g, 1.4 mmol) and triethylamine (2 mL) in THF (10 mL), substituted benzoyl chloride (1.5 mmol) was added dropwise. The mixture was stirred at room temperature for 3h. The THF was removed, and the target compounds **3a–3p** were given by column chromatography. The structure of the pure title product was confirmed by ^1^ H NMR, ^13^C NMR and HRMS. The spectra are given as Appendix A.

(*S*)-2-(2-chloronicotinamido)propyl 4-ethylbenzoate (**3a**) 

White solid, yield 43.4%, m.p. 77–79 °C; ^1^H NMR (CDCl_3_, 500 MHz), δ: 1.26 (t, *J* = 6.0 Hz, 3H, CH_3_), 1.40 (d, *J* = 5.7Hz, 3H, CH_3_), 2.69–2.73 (m, 2H, CH_2_), 4.43–4.45 (m, 2H, CH_2_), 4.62–4.66 (m, 1H, CH), 6.77 (d, *J* = 6.4 Hz, 1H, NH), 7.27 (d, *J* = 6.5 Hz, 2H, Ph), 7.30–7.33 (m, 1H, Py), 7.96 (d, *J* = 6.5 Hz, 2H, Ph), 8.01–8.03 (m, 1H, Py), 8.43–8.44 (m, 1H, Py); ^13^C NMR (CDCl_3_, 101 MHz) δ: 15.18, 17.21, 28.95, 45.64, 66.87, 122.72, 127.01, 128.01(2C, Ph), 129.78(2C, Ph), 131.28, 139.53, 147.06, 150.26, 150.89, 164.27, 166.63; HRMS (ESI) for C_18_H_19_ClN_2_O_3_ *m*/*z*: Calculated, 347.1157, Found, 347.1153 [M + H]^+^.

(*S*)-2-(2-chloronicotinamido)propyl 3,5-dimethylbenzoate (**3b**) 

White solid, yield 57.4%, m.p. 100–103 °C; ^1^H NMR (CDCl_3_, 500 MHz), δ: 1.41 (d, *J* = 5.6 Hz, 3H, CH_3_), 2.37 (s, 6H, CH_3_), 4.41–4.46 (m, 2H, CH_2_), 4.61–4.66 (m, 1H, CH), 6.79 (d, *J* = 6.3 Hz, 1H, NH), 7.21 (s, 1H, Ph), 7.32–7.34 (m, 1H, Py), 7.66 (s, 2H, Ph), 8.06–8.07 (m, 1H, Py), 8.44–8.46 (m, 1H, Py); ^13^C NMR (CDCl_3_, 101 MHz) δ:17.28, 21.15, 45.73, 66.86, 73.53, 122.78, 127.35(2C, Ph), 129.42, 131.14, 134.99, 138.18(2C, Ph), 139.79, 147.86, 150.99, 164.13, 166.91; HRMS (ESI) for C_18_H_19_ClN_2_O_3_ *m*/*z*: Calculated, 347.1157, Found, 347.1155 [M + H]^+^.

(*S*)-2-(2-chloronicotinamido)propyl 2-chlorobenzoate (**3c**) 

White solid, yield 57.6%, m.p. 77–79 °C; ^1^H NMR (CDCl_3_, 500 MHz), δ: 1.42 (d, *J* = 5.5 Hz, 3H, CH_3_), 4.45–4.47 (m, 2H, CH_2_), 4.64–4.68 (m, 1H, CH), 6.75 (d, *J* = 6.4 Hz, 1H, NH), 7.30–7.32 (m, 1H, Py), 7.32–7.35 (m, 1H, Ph), 7.42–7.46 (m, 2H, Ph), 7.86–7.87 (m, 1H, Ph), 8.01–8.03 (m, 1H, Py), 8.42–8.44 (m, 1H, Py); ^13^C NMR (CDCl_3_, 101 MHz) δ:17.35, 45.31, 67.67, 122.70, 126.75, 129.51, 131.14, 131.27, 131.69, 132.93, 133.57, 139.45, 147.06, 150.93, 164.29, 165.67; HRMS (ESI) for C_16_H_14_Cl_2_N_2_O_3_ *m*/*z*: Calculated, 353.0454, Found, 353.0456 [M + H]^+^.

(S)-2-(2-chloronicotinamido)propyl 3-chlorobenzoate (**3d**) 

White solid, yield 47.9%, m.p. 126–128 °C; ^1^H NMR (CDCl_3_, 500 MHz), δ: 1.42 (d, *J* = 5.4 Hz, 3H, CH_3_), 4.44–4.48 (m, 2H, CH_2_), 4.63–4.67 (m, 1H, CH), 6.71 (d, *J* = 6.4 Hz, 1H, NH), 7.32–7.35 (m, 1H, Py), 7.39–7.42 (m, 1H, Ph), 7.55–7.57 (m, 1H, Ph), 7.93–7.95 (m, 1H, Ph), 8.01–8.02 (m, 1H, Ph), 8.03–8.04 (m, 1H, Py), 8.44–8.45 (m, 1H, Py); ^13^C NMR (CDCl_3_, 125 MHz) δ: 17.19, 45.50, 67.43, 122.81, 127.79, 129.71, 129.86, 131.14, 131.35, 133.34, 134.66, 139.68, 147.01, 151.03, 164.28, 165.33; HRMS (ESI) for C_16_H_14_Cl_2_N_2_O_3_ *m*/*z*: Calculated, 353.0454, Found, 353.0460 [M + H]^+^.

(*S*)-2-(2-chloronicotinamido)propyl 4-(trifluoromethyl)benzoate (**3e**) 

White solid, yield 54.6%, m.p. 104–107 °C; ^1^H NMR (CDCl_3_, 500 MHz), δ: 1.41 (d, *J* = 5.4 Hz, 3H, CH_3_), 4.45–4.49 (m, 2H, CH_2_), 4.64–4.69 (m, 1H, CH), 6.71 (d, *J* = 6.4 Hz, 1H, NH), 7.31–7.33 (m, 1H, Py), 7.71 (d, *J* = 6.6 Hz, 2H, Ph), 8.00–8.02 (m, 1H, Py), 8.16 (d, *J* = 6.5 Hz, 2H, Ph), 8.42–8.44 (m, 1H, Py); ^13^C NMR (CDCl_3_, 101 MHz) δ:17.17, 45.44, 67.59, 122.82(2C, Ph), 124.85(CF_3_, *J* = 272Hz), 125.48(q, *J* = 4Hz), 130.06(2C, Ph), 131.05, 132.81, 134.57, 139.68, 146.93, 151.07, 164.26, 165.31; HRMS (ESI) for C_17_H_14_ClF_3_N_2_O_3_ *m*/*z*: Calculated, 387.0718, Found, 387.0719 [M + H]^+^.

(*S*)-2-(2-chloronicotinamido)propyl 2,6-difluorobenzoate (**3f**) 

White solid, yield 67.3%, m.p. 61–63 °C; ^1^H NMR (CDCl_3_, 500 MHz), δ: 1.41 (d, *J* = 5.6 Hz, 3H, CH_3_), 4.48–4.52 (m, 2H, CH_2_), 4.62–4.66 (m, 1H, CH), 6.69 (d, *J* = 6.4 Hz, 1H, NH), 6.97 (t, *J* = 6.6 Hz, 2H, Ph), 7.32–7.35 (m, 1H, Py), 7.42–7.47 (m, 1H, Ph), 8.05–8.07 (m, 1H, Py), 8.44–8.45 (m, 1H, Py); ^13^C NMR (CDCl_3_, 101 MHz) δ:17.19, 45.14, 67.85, 109.99, 112.06(d, *J* = 3Hz), 112.29(d, *J* = 3Hz), 122.72, 131.10, 133.35(t, *J* = 10.1Hz), 139.64, 147.12, 150.98, 162.07, 164.15; HRMS (ESI) for C_16_H_13_ClF_2_N_2_O_3_ *m*/*z*: Calculated, 355.0656, Found, 355.0660 [M + H]^+^.

(*S*)-2-(2-chloronicotinamido)propyl 2-ethoxybenzoate (**3g**) 

Waxy solid, yield 55.6%; ^1^H NMR (CDCl_3_, 500 MHz), δ: 1.36 (t, *J* = 5.6 Hz, 3H, CH_3_), 1.41 (d, *J* = 5.4 Hz, 3H, CH_3_), 4.06–4.10 (m, 2H, CH_2_), 4.40–4.45 (m, 2H, CH_2_), 4.59–4.64 (m, 1H, CH), 6.78 (d, *J* = 6.4 Hz, 1H, NH), 6.94–6.98 (m, 2H, Ph), 7.29–7.31 (m, 1H, Py), 7.44–7.47 (m, 2H, Ph), 7.79–7.81 (m, 2H, Ph), 7.99–8.01 (m, 1H, Py), 8.42–8.43 (m, 1H, Py); ^13^C NMR (CDCl_3_, 101 MHz) δ: 14.76, 17.42, 45.47, 64.31, 66.95, 112.97, 119.67, 120.10, 122.65, 131.46, 131.82, 133.77, 139.33, 147.15, 150.84, 158.43, 164.23, 166.57; HRMS (ESI) for C_18_H_19_ClN_2_O_4_ *m*/*z*: Calculated, 385.0926, Found, 385.0922 [M + Na]^+^.

(*S*)-2-(2-chloronicotinamido)propyl 3-(trifluoromethyl)benzoate (**3h**) 

White solid, yield 48.6%, m.p. 112–114 °C; ^1^H NMR (CDCl_3_, 500 MHz), δ: 1.41 (d, *J* = 5.6 Hz, 3H, CH_3_), 4.44–4.51 (m, 2H, CH_2_), 4.62–4.68 (m, 1H, CH), 6.74 (d, *J* = 6.4 Hz, 1H, NH), 7.31–7.33 (m, 1H, Py), 7.60 (t, *J* = 6.2 Hz, 1H, Ph), 7.83 (d, *J* = 6.2 Hz, 1H, Ph), 8.01–8.03 (m, 1H, Py), 8.23 (d, *J* = 6.3 Hz, 1H, Ph), 8.30 (s, 1H, Ph), 8.42–8.44 (m, 1H, Py); ^13^C NMR (CDCl_3_, 101 MHz) δ: 17.15, 45.46, 67.57, 122.80, 124.88(CF_3_, *J* = 270Hz), 126.52(q, *J* = 4Hz), 129.22, 129.79, 130.48, 131.04, 131.33, 132.85, 139.71, 146.94, 151.05, 164.27, 165.20; HRMS (ESI) for C_17_H_14_ClF_3_N_2_O_3_ *m*/*z*: Calculated, 387.0718, Found, 387.0732 [M + H]^+^.

(*S*)-2-(2-chloronicotinamido)propyl 2-methylbenzoate (**3i**) 

White solid, yield 41.7%, m.p.92–94 °C; ^1^H NMR (CDCl_3_, 500 MHz), δ: 1.42 (d, *J* = 5.4 Hz, 3H, CH_3_), 2.61 (s, 3H, CH_3_), 4.41–4.47 (m, 2H, CH_2_), 4.62–4.68 (m, 1H, CH), 6.71 (d, *J* = 6.2 Hz, 1H, NH), 7.25–7.28 (m, 2H, Ph), 7.32–7.35 (m, 1H, Py), 7.41–7.44 (m, 1H, Ph), 7.93–7.95 (m, 1H, Ph), 8.04–8.06 (m, 1H, Py), 8.45–8.46 (m, 1H, Py); ^13^C NMR (CDCl_3_, 101 MHz) δ:17.36, 21.86, 45.62, 66.83, 122.76, 125.81, 130.60, 131.13, 131.83, 132.36, 138.06, 139.70, 140.47, 147.03, 151.00, 164.16, 167.36; HRMS (ESI) for C_17_H_17_ClN_2_O_3_ *m*/*z*: Calculated, 333.1000, Found, 333.1009 [M + H]^+^.

(*S*)-2-(2-chloronicotinamido)propyl 2-(trifluoromethyl)benzoate (**3j**) 

Oil, yield 57.9%; ^1^H NMR (CDCl_3_, 500 MHz), δ: 1.31 (d, *J* = 5.4 Hz, 3H, CH_3_), 4.35–4.42 (m, 2H, CH_2_), 4.50–4.57 (m, 1H, CH), 6.85 (d, *J* = 6.5 Hz, 1H, NH), 7.20–7.23 (m, 1H, Py), 7.56–7.59 (m, 2H, Ph), 7.68–7.70 (m, 1H, Ph), 7.75–7.79 (m, 1H, Ph), 7.84–7.86 (m, 1H, Py), 8.31–8.32 (m, 1H, Py); ^13^C NMR (CDCl_3_, 101 MHz) δ: 17.09, 45.20, 68.22, 109.98, 121.98, 122.70, 124.70(CF_3_, *J* = 271Hz), 126.65(q, *J* = 5Hz), 130.50, 131.22, 131.52, 131.93, 139.39, 147.07, 150.92, 164.34, 166.74; HRMS (ESI) for C_17_H_14_ClF_3_N_2_O_3_ *m*/*z*: Calculated, 387.0718, Found, 387.0720 [M + H]^+^.

(*S*)-2-(2-chloronicotinamido)propyl 2-chloronicotinate (**3k**) 

White solid, yield 65.1%, m.p.106–108 °C; ^1^H NMR (CDCl_3_, 500 MHz), δ: 1.43 (d, *J* = 5.4 Hz, 3H, CH_3_), 4.46–4.52 (m, 2H, CH_2_), 4.64–4.70 (m, 1H, CH), 6.67 (d, *J* = 6.5 Hz, 1H, NH), 7.32–7.37 (m, 2H, Py), 8.02–8.04 (m, 1H, Py), 8.22–8.23 (m, 1H, Py), 8.43–8.45 (m, 1H, Py), 8.52–8.53 (m, 1H, Py); ^13^C NMR (CDCl_3_, 101 MHz) δ:17.30, 45.22, 68.23, 122.26, 122.78, 126.43, 131.05, 139.57, 140.69, 146.98, 149.84, 151.06, 152.17, 164.29, 164.50; HRMS (ESI) for C_15_H_13_Cl_2_N_3_O_3_ *m*/*z*: Calculated, 354.0407, Found, 354.0409 [M + H]^+^.

(*S*)-2-(2-chloronicotinamido)propyl 4-chlorobenzoate (**3l**) 

White solid, yield 52.8%, m.p. 89–91 °C; ^1^H NMR (CDCl_3_, 500 MHz), δ: 1.31 (d, *J* = 5.4 Hz, 3H, CH_3_), 4.45–4.51 (m, 2H, CH_2_), 4.62–4.69 (m, 1H, CH), 6.68 (d, *J* = 6.5 Hz, 1H, NH), 7.27–7.30 (m, 2H, Ph), 7.32–7.34 (m, 1H, Py), 7.60–7.62 (m, 1H, Ph), 7.69–7.71 (m, 1H, Ph), 8.02–8.04 (m, 1H, Py), 8.44–8.45 (m, 1H, Py); ^13^C NMR (CDCl_3_, 101 MHz) δ:17.32, 45.28, 68.00, 122.77, 127.28, 129.29(2C, Ph), 131.08, 132.32, 133.46(2C, Ph), 139.59, 147.02, 151.04, 164.24, 165.34;HRMS (ESI) for C_16_H_14_Cl_2_N_2_O_3_ *m*/*z*: Calculated, 353.0454, Found, 353.0450 [M + H]^+^.

(*S*)-2-(2-chloronicotinamido)propyl 2,3-dichlorobenzoate (**3m**) 

White solid, yield 47.9%, m.p. 89–91 °C; ^1^H NMR (CDCl_3_, 500 MHz), δ: 1.42 (d, *J* = 5.4 Hz, 3H, CH_3_), 4.45–4.51 (m, 2H, CH_2_), 4.62–4.69 (m, 1H, CH), 6.66 (d, *J* = 6.5 Hz, 1H, NH), 7.27–7.30 (m, 1H, Ph), 7.32–7.35 (m, 1H, Py), 7.61–7.63 (m, 1H, Ph), 7.69–7.71 (m, 1H, Ph), 8.03–8.05 (m, 1H, Py), 8.44–8.46 (m, 1H, Py); ^13^C NMR (CDCl_3_, 125 MHz) δ: 17.32, 45.29, 68.00, 122.77, 127.30, 129.31, 131.13, 131.53, 132.34, 133.47, 134.69, 139.56, 147.04, 151.04, 164.27, 165.35; HRMS (ESI) for C_16_H_13_Cl_3_N_2_O_3_ *m*/*z*: Calculated, 387.0065, Found, 387.0071 [M + H]^+^.

(*S*)-2-(2-chloronicotinamido)propyl 2,4-dichlorobenzoate (**3n**) 

White solid, yield 48.3%, m.p. 116–118 °C; ^1^H NMR (CDCl_3_, 500 MHz), δ: 1.42 (d, *J* = 5.6 Hz, 3H, CH_3_), 4.44–4.49 (m, 2H, CH_2_), 4.63–4.70 (m, 1H, CH), 6.66 (d, *J* = 6.5 Hz, 1H, NH), 7.32–7.36 (m, 2H, Ph, Py), 7.49 (s, 1H, Ph), 7.86 (d, *J* = 6.7 Hz, 1H, Ph), 8.04–8.06 (m, 1H, Py), 8.45–8.46 (m, 1H, Py); ^13^C NMR (CDCl_3_, 125 MHz) δ: 17.36, 45.29, 67.89, 122.79, 127.23, 127.76, 131.11, 131.13, 132.84, 134.82, 138.81, 139.65, 147.04, 151.06, 164.23, 164.78; HRMS (ESI) for C_16_H_13_Cl_3_N_2_O_3_ *m*/*z*: Calculated, 387.0065, Found, 387.0070 [M + H]^+^.

(*S*)-2-(2-chloronicotinamido)propyl 4-methoxybenzoate (**3o**) 

White solid, yield 67.6%, m.p. 108–110 °C; ^1^H NMR (CDCl_3_, 500 MHz), δ: 1.41 (d, *J* = 5.4 Hz, 3H, CH_3_), 4.39–4.46 (m, 2H, CH_2_), 4.60–4.67 (m, 1H, CH), 6.73 (d, *J* = 6.4 Hz, 1H, NH), 6.93 (d, *J* = 6.6 Hz, 2H, Ph), 7.32–7.34 (m, 1H, Py), 8.01 (d, *J* = 6.7 Hz, 2H, Ph), 8.02–8.04 (m, 1H, Py), 8.44–8.45 (m, 1H, Py); ^13^C NMR (CDCl_3_, 101 MHz) δ:17.22, 45.70, 55.45, 66.78, 113.74(2C, Ph), 121.91, 122.74, 131.26, 131.72(2C, Ph), 139.59, 147.05, 150.93, 163.62, 164.24, 166.32; HRMS (ESI) for C_17_H_17_ClN_2_O_4_ *m*/*z*: Calculated, 349.0950, Found, 349.0961 [M + H]^+^.

(*S*)-2-(2-chloronicotinamido)propyl 2-nitrobenzoate (**3p**) 

oil, yield 63.6%; ^1^H NMR (CDCl_3_, 500 MHz), δ: 1.29 (d, *J* = 5.4 Hz, 3H, CH_3_), 4.28–4.31 (m, H, CH), 4.44–4.53 (m, 2H, CH_2_), 6.78 (d, *J* = 6.4 Hz, 1H, NH), 7.22–7.24 (m, 1H, Py), 7.60–7.66 (m, 2H, Ph), 7.75–7.77 (m, 1H, Ph), 7.79–7.81 (m, 1H, Ph), 7.85–7.88 (m, 1H, Py), 8.31–8.33 (m, 1H, Py); ^13^C NMR (CDCl_3_, 101 MHz) δ:17.01, 45.16, 68.51, 122.62, 123.70, 126.62, 130.36, 131.41, 132.26, 132.91, 139.14, 139.17, 147.24, 150.81, 164.56, 165.00; HRMS (ESI) for C_16_H_14_ClN_3_O_5_ *m*/*z*: Calculated, 364.0695, Found, 364.0700 [M + H]^+^.

### 3.3. Fungicide Bioassays

The fungicidal activity of title chiral niacinamide derivatives **3a**~**3p** were tested in vitro against *F. oxysporum* (FO), *C. arachidicola* (CA), *B.berengriana* (BB), *A. solani* (AS), *G. zeae* (GZ), *S. sclerotiorum* (SS), *B. cinerea* (BC), *R. solani* (RS), *P. infestans* (PI) and *P. capsici* (PC), and their relative percent inhibition (%) was determined using the mycelium growth rate method according to the previous work [27,28]. Fluxapyroxad was used as the positive control. Each compound was dissolved in DMSO with 1% tween to prepare the 10 g/L stock solution. The ten fungal species were inoculated into a Petri dish containing 50 μg/mL stock solution and incubated in a 25 °C biochemical incubator in darkness. The solvent DMSO with 1% tween was used as a blank assay. The fungicidal effect was determined 3 days later. Each treatment (species and compound) was repeated three times. The inhibition of compounds compared to the blank assay was calculated via the following equation:Inhibition (%) = (CK − AI)/CK × 100% 
where CK is the average diameter of mycelia in the blank test and AI is the average diameter of mycelia in the presence of those compounds. The EC_50_ values were also calculated.

### 3.4. Herbicide Bioassays

The bioassay method of Dayan et al. [29] was used. Briefly, seeds of lettuce (Latuca sativa—cv. Iceberg A Crisphead from Burpee Seeds of Warminster, PA, USA) and bentrass (Agrostis stolonifera—Penncross variety obtained from Turf-Seed, Inc. of Hubbard, OR, USA) were surface sterilized with a 0.5% to 1% (*v*/*v*) sodium hypochlorite solution for approximately 10 min, rinsed with deionized water and dried in a sterile environment. A filter paper disk (Whatman Grade 1, 1.5 cm) was placed in each well of a 24-well plate. The control wells contained 200 μL of deionized water. The control + solvent well contained 180 μL of water and 20 μL of the solvent. All sample wells contained 180 μL of water and 20 μL of the appropriate dilution of the sample. Water was pipetted into the well before the sample or solvent. Test samples were dissolved in acetone and the final concentration of acetone in the wells was 10%. For the bioassay five lettuce seeds or 20 mg of bentgrass seeds were placed in each well before sealing the plate with Parafilm. The plates were incubated for seven days in a Percival Scientific (Perry, IA, USA) CU-36L5 incubator under continuous light conditions at 26 °C and 120 μmol∙s^−1^∙m^−2^ average photosynthetically active radiation (PAR). A qualitative estimate of phytotoxicity was made by assigning a rating of 0 for no effect (sample well plants looked identical to the control plus solvent well plants; seeds had germinated and resulting seedlings had grown normally), 1 less than 20% inhibited germination, 2 for about 20–40% germination inhibition, 3 for about 40–60% germination inhibition, 4 for more than 60% germination inhibition, and 5 for no germination of the seeds. Each experiment was repeated.

### 3.5. Molecular Docking

Modeled protein succinate dehydrogenase (SDH) and compound **3i** and positive control fluxapyroxad were prepared using Discovery Studio 3.5 software according to our previous work [30,31]. All docking calculations were performed using the Discovery Studio 3.5 software package and energy minimization was carried out by CHARMm [32] force field using the ligand partial charge method CFF (Consistent Force Field). Minimization was carried out until an energy gradient of 0.01 was reached. The CDOCKER was used for docking of the two compounds. The X-ray crystal structure of succinate dehydrogenase (SDH, 2FBW [33]) was downloaded from the protein data bank, which was used and refined with CHARMm in DS 3.5 at physiological pH. Consequently, compound **3i** and the positive control fluxapyroxad were docked into the active site; ten conformations of each compound were obtained through CDOCKER based on the docking score after energy minimization using the smart minimizer method, which begins with the steepest descent method, followed by the conjugate gradient method. For each final pose, the CHARMm energy and the interaction energy are calculated. The poses are sorted by CHARMm energy, and the tentop scoring poses are retained [34]. The binding modes were validated by interactions between the candidate molecules and active site residues.

## 4. Conclusions

In conclusion, a series of *(S)-*2-(2-chloronicotinamido)propyl benzoate compounds was synthesized using 2-chloronicotinic acid as starting material. These compounds were introduced with a flexible, chiral chain. The fungicidal and herbicidal activity of the synthesized compounds against ten fungi and two plant species were tested, and some of the compounds showed good fungicidal activity against *B. berengriana* and *S. sclerotiorum*, and low phytotoxic against the monocot and dicot, allowing them to be used in plant fungal protection programs. Molecular binding studies indicate that the most active compound **3i** is a probably a SDH inhibitor. These structures can be further optimized for discovery of new fungicides.

## Data Availability

Data are contained within the article or Appendix A.

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
