# Peer review of "Synthesis and Pesticidal Activity of New Niacinamide Derivatives Containing a Flexible, Chiral Chain"

_molecules, 2022, doi:10.3390/molecules28010047_

Round 1

Reviewer 1 Report

In this manuscript, «Synthesis and Pesticidal Activity of New Niacinamide Derivatives Containing a Flexible, Chiral Chain», the authors present the synthesis, new niacinamide derivatives, containing chiral flexible chains, and test of fungicidal and herbicidal activities of these compounds. In addition, the authors also present molecular docking. This article is a logical continuation of the research team of authors. The article is well-written, all sections logically follow each other. The paper is of high importance, since the synthesis of new effective and non-toxic pesticides is a very important task in modern conditions of life.

I believe that this article is suitable for publication in Molecules. However, I have a number of comments:

1. Line 31 contains an extra dot after the word «crops».

2. Line 52: The «N» in the IUPAC name of the compound should be italicized.

3. Lines 53, 54, 57: There are abbreviations of words without explanations (sp. and spp.).

4. Lines 56: It is required to put a dot after the abbreviation «et al».

5. Lines 68, 69: 1 and 13 near the atomic signs must be in upper case.

6. Lines 82, 83: The «S» in the IUPAC name of the compound should be italicized.

7. Line 88: The line contains an abbreviation without decryption (TLC).

8. Line 89: Scheme 1 contains an explanation of the radicals at the bottom in small font. I would like to increase the font size.

9. Line 99: The «S» in the IUPAC name of the compound should be italicized.

10. Lines 99-101: The names of fungal strains are given immediately in abbreviated form. Perhaps you should first give in full format.

11. Line 101: The curly brace should be replaced with a regular one.

12. Line 102: The reference to the previous work of the authors is required.

13. Line 103: The table contains many abbreviations. However, there is only one transcript in the text. Perhaps some clarification should be added to the table legend. CK is the control sample. But this is not mentioned anywhere in the text of the manuscript.

14. Lines 131, 132: The «S» in the IUPAC name of the compound should be italicized.

15. Line 141: In column R it is not entirely correct to indicate Ph if there is a substitution. It is more correct to write C6H4 for one substituent and C6H3 for two substituents.

16. Line 326: The «S» in the IUPAC name of the compound should be italicized.

17. Line 347 contains information about the availability of the Supplementary Materials. However, for some reason, I, as a reviewer, was not provided with this file. Does it exist?

18. There is also not enough information about molecular docking. I would like to evaluate and compare the binding energy of 3i and the reference compound.

19. Section 3. Materials and Methods does not contain information about molecular docking. It is required to add information about software, protein, etc.

Author Response

see attachement

Reviewer 2 Report

Review of the manuscript

ID molecules-2060161

"Synthesis and Pesticidal Activity of New Niacinamide Derivatives Containing a Flexible, Chiral Chain" Zhe-Cheng Wei, Qiao Wang, Li-Jing Min, Joanna Bajsa-Hirschel, Charles L. Cantrell, Liang Han, Cheng-Xia Tan, Jian-Quan Weng, Yu-Xin Li, Na-Bo Sun, Stephen O. Duke and Xing-Hai Liu

This manuscript presents a three-stage synthesis of nicotinic acid derivatives, including a chiral amide fragment and an ester of substituted benzoate or nicotinate. Fungicidal and herbicidal activity was studied for 16 such new derivatives, compounds were found that are not inferior to fluxapyroxad.

There is a fundamental remark to this work. The article presents 13C NMR spectra only for 3d, m, n compounds, they are absent for other compounds. The authors write (l.185): "The spectra are given as supporting information.", however, they did not attach any additional files to the article. It is recommended to give the data of the 13C spectra in the text of the article. In this form, as it is now, the work cannot be published in this volume. It is necessary either to provide data from NMR 13C spectra, or to remove compounds for which there are no NMR 13C spectra.

A thorough revision of the English language style is required before publication.

In addition, there are a number of inaccuracies in the work that the authors can correct:

l.76, l.131, l.133, l.326 - Compounds 3 are benzoates, not propionate. Compound 3k is nicotinate.

l.77- Usually the SOCl2 compound is called "thionyl chloride" and not "sulfoxide chloride". Or is it about "sulfuryl chloride", SO2Cl2?

l.82 - "acyl chloride" (must be added) "1".

l.84 - What is (4)? If it is a literary reference, then write [4].

l.85 - what by-product is meant?

l.88- Instead of "acyl chloride" it is better to write "substituted benzoyl chlorides (and nicotinyl chloride 3k)".

l.90 - The signature to scheme 1 is uninformative.

l.92 - Compounds 3 do not have an amino group, only an amide group.

l.93 - "The pyridine ring and the benzene ring are located from 6-9 ppm." Slang expression, needs to be corrected.

To Table 1, it is necessary to explain which microorganisms are encrypted (CA, BB, etc.), and specify the full, not abbreviated species name. This can be done at the first mention of them in the text of the work.

How was phytotoxicity determined in Table 3 in terms of "1-4"?

l.333 - Docking only shows that the compound may be a potential inhibitor, and not a real one.

Not all literary references have a DOI specified.

Author Response

see attachement

Round 2

Reviewer 2 Report

Review of the manuscript

ID molecules-2060161-v2

"Synthesis and Pesticidal Activity of New Niacinamide Derivatives Containing a Flexible, Chiral Chain" Zhe-Cheng Wei, Qiao Wang, Li-Jing Min, Joanna Bajsa-Hirschel, Charles L. Cantrell, Liang Han, Cheng-Xia Tan, Jian-Quan Weng, Yu-Xin Li, Na-Bo Sun, Stephen O. Duke and Xing-Hai Liu

This manuscript is reviewed by the reviewer again. All the comments identified earlier have been corrected by the authors. In the course of editing the work, the authors inserted a new table under the number 2 (l.176), but this number should be 4. It is quite possible that these new data can be presented in text form, and not in a table, since there are not many of them.

The article presents new and interesting data on the fungicidal and herbicidal activity of new nicotinic acid derivatives. The work should be interesting to a wide range of readers and deserves publication.